# Deep-learning estimation of second-generation pharmacokinetic-model parameters in DCE-MRI

Denisa Hývlová
*Institute of Scientific Instruments*
*Czech Academy of Sciences*
and
*Faculty of Electrical Engineering*
*Brno University of Technology*
Brno, Czechia
Email: hyvlova@isibrno.cz

Radovan Jiřík
*Institute of Scientific Instruments*
*Czech Academy of Sciences*
Brno, Czechia
Email: jirik@isibrno.cz

Jiří Vitouš
*Institute of Scientific Instruments*
*Czech Academy of Sciences*
and
*Faculty of Electrical Engineering*
*Brno University of Technology*
Brno, Czechia
Email: vitous@isibrno.cz

*Abstract*—Dynamic contrast-enhanced magnetic resonance imaging (DCE-MRI) is a promising method for the evaluation of tissue perfusion. Current standard is fitting of a pharmacokinetic model to the acquired signals. Most commonly, first generation models are used (Tofts, extended Tofts model) providing stable results, however, only a limited set of parameters. Second generation models allow estimation of a larger parameter set, thus a more complete description of the perfusion state, however, they require high data quality and their application is more computationally demanding. Overall, the lack of standardization of DCE-MRI, its computational time and reliability hinders its routine clinical application. Deep learning methods allow fast parameter estimation and bring new possibilities into this field. In this study, we have explored the application of a convolutional neural network for the prediction of second-generation model parameters. The network was tested for different noise levels and sampling periods on a simulated dataset, and the results were validated on a real preclinical dataset. The proposed method provided more stable and robust results compared to the conventional model fitting.

*Index Terms*—Deep learning, MRI, perfusion imaging

## I. INTRODUCTION

Dynamic contrast-enhanced magnetic resonance imaging (DCE-MRI) is a promising method for the evaluation of perfusion and permeability changes in tissues. These changes can be valuable for diagnosing and monitoring treatment effects, particularly in oncology, neurology, and cardiology. [1] For example, when monitoring tumor response to treatment, morphological changes can typically be observed within months, while perfusion changes emerge within days, bringing very early insights into the treatment effects. [2]

In DCE-MRI, a paramagnetic contrast agent is administered and its passage through tissue is tracked in a series of T1-weighted MR images. The acquired image sequence is converted to a contrast-agent concentration image sequence using a pre-contrast sequence and a selected T1-mapping method. The contrast-agent concentration in a tissue $c(t)$ can be defined

as a convolution of contrast-agent concentration in plasma of the feeding artery $c_p(t)$, also known as the arterial input function (AIF), and a so called tissue residue function $H(t)$,

$$c(t) = (c_p * H)(t). \tag{1}$$

The tissue residue function $H(t)$ is modeled with a pharmacokinetic (PK) model, parametrized with a corresponding set of perfusion and permeability parameters. The AIF can be measured in a large artery from the acquired DCE-MRI dataset, or population-based model can be selected. The only unknown of the equation is the $H(t)$ function with its parameters; therefore, by fitting of each voxel's concentration curve with a selected PK model, the sought perfusion and permeability parameters can be obtained. [3]

The most commonly used PK models are the 1st-generation models, describing the tissue with 2–3 parameters (e.g. Patlak, Tofts, or extended Tofts model). More realistic description is provided by the 2nd-generation models, which are defined by 4–5 parameters. Their usage usually requires higher temporal resolution of the acquired image sequence, achievable e.g. with compressed sensing approaches, reconstructing image data from undersampled k-space. [4]

A common approach to fitting of a PK model to the concentration curves is the non-linear least-squares (NLLS) technique. For large datasets, this procedure can be very time-consuming, taking tens of minutes to hours. This might be an obstacle for the application of DCE-MRI in clinical practice, where fast computation of results is highly desired to evaluate patient data as quickly as possible. The computational time increases with the number of model parameters; therefore, it is an issue especially for the 2nd-generation models. [1]

With the advances in artificial intelligence, alternative approaches to NLLS techniques have also been introduced, in particular the training of neural networks (NNs) to predict the perfusion and permeability parameters from the DCE-MRI data. Although the training of a NN is time-consuming, the prediction is significantly faster than the NLLS techniques. Also, there might be other advantages in terms of possibly

This work was supported by the Czech Science Foundation grant (GA22-10953S). All MR experiments were carried out at the ISI-MR facility of the Czech-BioImaging infrastructure, supported by grant LM2023050 of the MEYS CR.

lower sensitivity to noise and to the problem of false local optima.

The training of NNs to predict the perfusion and permeability parameters can occur in either a supervised or self-supervised manner. In the supervised approach, the NN is trained using a loss function that measures the distance between the DL-based perfusion parameters and those obtained from traditional NLLS-based methods. Alternatively, when training on simulated data, the parameters used for the generation of DCE-MRI data can be used as labels. [5]–[10]

In the self-supervised approach, the NN is trained based on the quality of fit of the concentration curves generated from the DL-estimated perfusion parameters compared to the original concentration curves. This approach requires selection of a PK model for generation of the DL-based concentration curves for the loss function, but it should result in a more robust parameter estimation. [11]–[13]

The architectures can be categorized into two main groups — temporal and spatio-temporal. Temporal networks learn the voxelwise parameter estimation, commonly employing recurrent architectures [5], [13] or convolutional neural networks (CNNs) [14]. On the other hand, spatio-temporal networks incorporate spatial information from the local neighborhood, similarly to spatial regularization of the NLLS approach [15], improving the parameter estimates. Presently, CNNs in various configurations dominate the employed architectures [6]–[10].

DL approaches also diverge in handling of the input data. Apart from the conventional preprocessing of the DCE data, that is the conversion of MR signal intensity to concentration, with the concentration curves supplied as the model input, the preprocessing can be bypassed, with the image sequence fed directly into the model. [7]–[9], [14]

The current studies vary in all the above-mentioned aspects, but almost exclusively rely on the 1st-generation PK models. To our knowledge, there is only one study focused on the application of a 2nd-generation model [12], where a voxelwise, physics-informed architecture was implemented. Therefore, because this field remains mainly unexplored, our aim is to design a convolutional model for prediction of the 2nd-generation model-parameters and to estimate its accuracy and precision in comparison with NLLS techniques under different conditions.

## II. MATERIALS AND METHODS

### A. Model

For the task, we designed a CNN architecture depicted in Fig. 1. The input of the network are concentration curves, the output is a set of parameters $\{F_p, v_p, v_e, \mathrm{PS}, \mathrm{BAT}\}$, where $F_p$ is plasma flow, $v_p$ is plasma volume, $v_e$ is extravascular extracellular volume, PS is the permeability–surface-area product, and BAT is the bolus arrival time. This parameter set could be output by the two-compartment exchange (2CX) model [3], which was chosen for subsequent processing and analysis in this study as one of the most common 2nd-generation PK models.

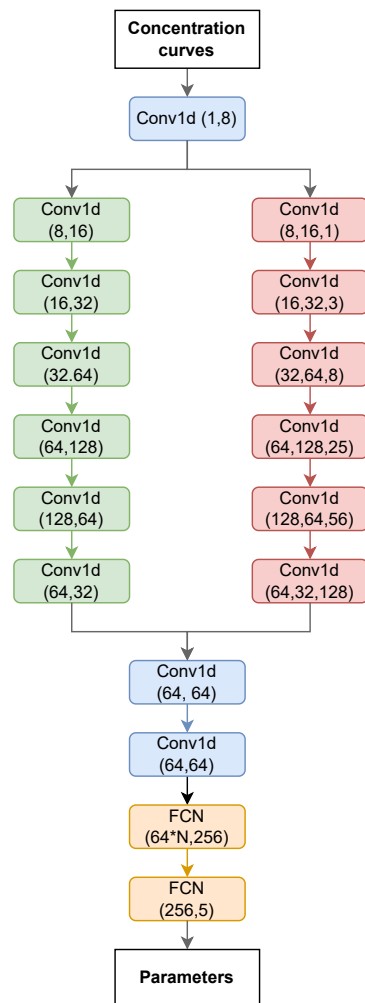

Fig. 1: Designed CNN architecture. Feature extraction and merging is shown in blue, local pathway in green, global pathway in red. Fully connected network extracting final perfusion and permeability parameters is depicted in yellow. Numbers refer to the number of input channels, output channels, and, for the global pathway, the dilatation factor. N is the number of timepoints of the input curves.

At first, low-level features are extracted from the concentration curves using a 1D convolutional layer. The features are passed in parallel into a local and global pathway, which is known to provide good results for the estimation of parameters from DCE-MRI data. [6], [8], [14] Each pathway consists of 6 convolutional layers. The local pathway is processing the signals with a stride of 1, the global pathway uses dilated convolution with increasing dilation factor to extract long-term information. Zero-padding is applied in both patways to ensure matching sizes of the input and output signals. Outputs of both pathways are then concatenated and merged by two more 1D convolutional layers. All the convolutional layers are followed by a ReLU activation function. Finally, the features are processed by two fully-connected layers; the first is followed by a leaky ReLU activation function, the second

is followed by a sigmoid activation function.

The output parameters are scaled to the expected physiological range; that is for $F_p$ to [0, 5] mL/min/mL and for PS to [0, 2] mL/min/mL, the $v_p$ and $v_e$ are kept to [0, 1] mL/mL. The BAT is estimated in a given range around the precontrast scan time, which is commonly manually or automatically set during the processing of DCE-MRI data. The range was set between -0.1 min and +0.2 min. The precontrast scan time, passed as an input information, is added to the estimated value to obtain the BAT.

### B. Loss function and hyperparameters

The network was trained using a self-supervised approach; that means, the predicted parameters were used for the generation of the concetration curves, and the loss function was defined as a normalized mean square error between the original and predicted concentration curves,

$$\text{NMSE} = \frac{1}{N} \sum_{n=1}^{N} \frac{\sum_{i=1}^{I}(c(t_i) - \hat{c}(t_i))^2}{\sum_{i=1}^{I} c(t_i)^2}, \quad (2)$$

where $t_i$ are the sampling time points of the concentration curves, $I$ is the number of the curves' samples, and N is the number of samples in a batch.

For the generation of the concentration curves, an implementation of the 2CX model was used. The curves were simulated with a three gamma-variate function (3GVF) AIF, designed to match the vascular-system dynamics of small animals. [16]. The parameters of the AIF were found using blind-deconvolution AIF estimation from a real dataset used later for the testing of the model.

Based on hyperparameter optimization, batch size was chosen as 512 and learning rate as 0.0001. For the training, Adam optimizer combined with a scheduler with the patience of 3 epochs and factor 0.1 was chosen (number of epochs with no improvement after which learning rate was reduced). The network was trained for 50 epochs in total.

### C. Training and validation data

For the training of the CNN, simulated concentration curves with different levels of noise were used. The curves were generated with the AIF mentioned above and the 2CX model. The total of 500 000 curves with random parameters were simulated, with parameter ranges as follows: $F_p$ = [0.08, 5] mL/min/mL, $v_p$ = [0.0005, 0.1] mL/mL, $v_e$ = [0.01, 0.7] mL/mL, PS = [0.001, 2] mL/min/mL, and BAT = [0.4, 0.8] min. To represent plasma flow value distribution in tissues more accurately, we selected the plasma flow from a logarithmic random distribution. For all other parameters, uniform distributions were chosen. The sampling period $T_s$ was set to 1.2 s and the signal length to 256 seconds to match the testing data. To mimic the *in vivo* conditions, we added zero-mean Gaussian noise to the curves with the standard deviation ranging randomly from 0 to 0.1 with a uniform distribution, providing realistic levels of noise, as compared with our real DCE-MRI datasets. The final dataset was split to the training and validation set with a ratio of 0.8.

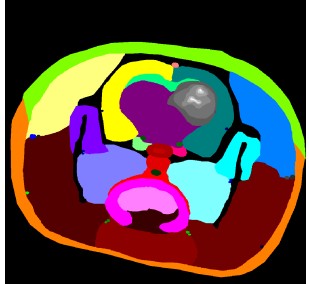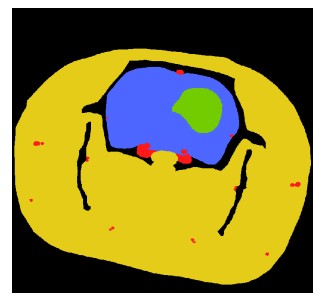

Fig. 2: Virtual phantom used for the generation of synthetic DCE-MRI data. Left – all segmented tissues, right – main regions: tumor (green), brain (blue), vessels (red), and muscles, skin and other tissues (yellow).

### D. Testing data

As the main application of DCE-MRI is the diagnosis of oncological diseases, the implemented architecture was tested on two tumor datasets – the first dataset was a synthetic DCE-MRI of a glioblastoma-bearing rat, the second one was an *in vivo* measurement of a mouse with a subcutaneously implanted tumor.

*1) Synthetic data:* In-house made simulation software Perf-Sim was used to generate realistic synthetic DCE-MRI data [17]. The input of the simulation is a virtual phantom composed of segmented tissue regions, based on real MR images. Each tissue region is assigned a corresponding set of perfusion parameters necessary for the simulation. The simulator generates the concentration curves using a predefined PK model and an AIF for each region and converts them to the MR signal intensity based on a model of the FLASH MR acquisition. A synthetic image sequence with a high temporal and spatial resolution is then created from the signal curves and the virtual phantom, and weighted by input coil sensitivities (measured using coils of a real MR scanner). Raw MR echo signals are generated by transforming the synthetic image sequence into the $k$-space. The simulator allows Cartesian, radial and rosette acquisition. Gaussian noise may be added to the echo signals. The software implements simulation of multi-flip-angle, multi-repetiton-time and IRLL pre-contrast sequences (scans needed to convert the MR image intensity to contrast-agent concentration).

A virtual phantom representing an axial slice of a rat's head with glioblastoma was used in this study, shown in Fig. 2. A compressed-sensing DCE-MRI acquisition was simulated with the following parameters: RF-spoiled FLASH sequence with radial golden angle (GA) sampling [4] acquired with a surface rat-brain four-channel coil, TR = 8 ms, TE = 1.445 ms, FA = 15°, 38400 projections, total acquisition time 256 seconds, and matrix size of the reconstructed images 128×128.

Three datasets were simulated: 2D without noise, 2D with realistic noise, and 3D with realistic noise (acquisition of 10 slices was simulated, i.e. image reconstruction was done from 10× more undersampled MR data). We set the standard deviation of Gaussian noise added to the echo signals as 0.022

to achieve a similar level of noise as in the original dataset, from which the perfusion parameters used as ground-truth in the software were estimated. The IRLL pre-contrast sequence [18] was simulated with the following parameters: $\tau = 8.2$ ms, $t_d = 12.1$ ms, $t_r = 8.2$ ms, FA = 3°, TE = 1.6762 ms, and 1500 projections per inversion.

The image sequences of both the dynamic and the pre-contrast data were reconstructed from the synthetic $k$-space data using the BART reconstruction toolbox [19]. The temporal sampling of the DCE sequence was set to 1.2 seconds. The reconstruction was performed with the total variation (TV) regularization in both the spatial and the temporal domains, with the weights set experimentally to 0.001 in the spatial domain and 0.0001 in the temporal domain so that the optimal results of the reconstruction and perfusion analysis were achieved. The IRLL image sequence was reconstructed also with TV regularization, with the weight set to 0.0004 for the spatial domain.

To test the ability of the CNN to handle various sampling periods, we reconstructed the 2D dataset with added noise also with the temporal sampling period of 2.4 s and 4.8 s. The regularization weights were set lower for the dataset with 4.8 s to prevent overregularization.

To obtain the concentration curves from the simulated MR images for the testing of the CNN and to calculate the NLLS perfusion maps, we processed the DCE-MRI sequences in the PerfLab software [20], fitting the sequences with the 2CX model in a voxelwise manner. The same AIF as for the simulation of training data was used for the deconvolution.

*2) Real data:* A tumor-bearing mouse was imaged with a 9.4T Bruker BioSpec USR 94/30 (Bruker BioSpin, Ettlingen, Germany) scanner and a surface eight-channel coil. A bolus of 0.2 mmol/kg Gadovist® (Bayer AG, Germany) contrast agent was administered into the tail vein after 45 seconds of acquisition using a linear infusion pump (Harvard Apparatus) with the injection speed of 1 mL/min. The experiments were approved by the National Animal Research Authority.

A single axial slice was imaged using a custom-made 2D GA multi-gradient echo sequence with parameters: TR/TE = 23/1.4, 3.4, 5.4, 7.7, 9.4, 11.5, 13.5, 15.5 ms, FA = 30°, acquisition time 15 min, and matrix size of the reconstructed image 128×128. From the multi-gradient echo sequence, only the first echo was picked for the analysis here as used commonly. Multi-echo data were used only in the blind-deconvolution AIF estimation. This gave more reliable results than for single echo [21]. 2D IR GA precontrast sequence was captured for the T10 estimation with TE = 1.46 ms, TR = 23 ms, and FA = 30°.

The images were reconstructed using the BART toolbox with $T_s = 1.2$ s. For the dynamic sequence, wavelet regularization was used in the spatial domain with the weight set to 0.001 and TV regularization was used in the temporal domain with the weight set to 0.005. The precontrast image sequence was reconstructed with TV regularization with the weight set to 0.001.

To obtain the concentration curves from the real MR images for the testing of the CNN and to calculate the NLLS perfusion maps, we processed the DCE-MRI sequence in the PerfLab software [20], with the same AIF as was used for the simulations and the 2CX model fitted in a vowelwise manner.

## III. RESULTS

### A. Synthetic data

The results obtained from the synthetic datasets with different noise levels are shown in Fig. 3. To improve the clarity of the final parameter maps, we omitted the brain region and the vessels from the analysis of all the simulated datasets, because the 2CX model is not suitable for these specific tissue types. In addition, we are not showing the BAT parameter, which does not hold any significant diagnostic information and is important mainly for the purpose of curve fitting.

For the noise-free dataset (2nd and 3rd column), both the NLLS and the CNN-based approaches provided similar results, corresponding with the ground truth. For the 2D dataset with added noise, the CNN outperformed the NLLS parameter estimation, which was clearly more sensitive to the present noise. For the 3D dataset with noise, the results had shown even more significant difference between the methods than for the 2D noisy data, with the NLLS method providing incorrect estimates predominantly in the muscles. The most noise-sensitive parameters to estimate were $F_p$ and $v_p$. The best results were obtained for the $v_e$ parameter; however, the CNN still outperformed the NLLS method for these parameters in the presence of noise.

The resulting maps in the region of interest, glioblastoma, are shown in detail in Fig. 4. In contrast to the overall noise-sensitivity of the NLLS method for the whole rat-head phantom, composed mainly from muscles, the estimated parameters in the glioblastoma were less influenced by noise for this method. The most imprecise parameter estimates were obtained for $F_p$. The remaining parameters were estimated predominantly correctly for the NLLS method, even for the noisy data. As for the whole phantom, the CNN provided stable results for all the parameters under all tested conditions.

The error between the parameter estimates and the ground truth was quantified using the mean absolute percentage error (MAPE), and with the standard deviation (SD) is summarized in Table I for the whole rat-head phantom and for the glioblastoma only. The results correspond with the differences observable from the parameter maps. For the whole phantom and noise-free data, the CNN achieved lower MAPE for $F_p$ and BAT, however, for the $v_p$, $v_e$ and PS, the MAPE was slightly higher. In the presence of noise, the CNN outperformed the NLLS and the MAPE was lower for all the parameters. The SD of the CNN estimates was lower for all parameters and datasets, as was expected from the perfusion maps.

For the glioblastoma only, the CNN achieved lower MAPE than the NLLS method for $F_p$ estimates in all datasets; for the remaining parameters, the deviations from the ground truth were comparable for both methods, with the difference between them being mostly less than a percent. The SD of the estimates followed a similar trend.

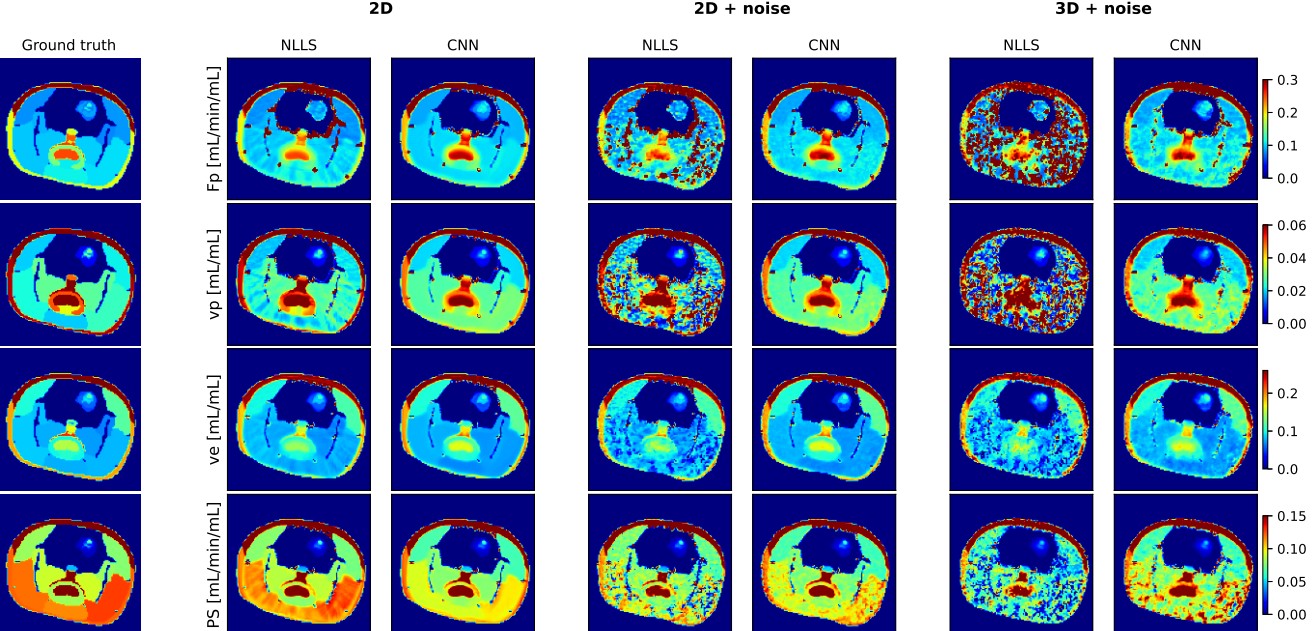

Fig. 3: Estimated parameters from the CNN and the NLLS method for 2D data without noise, 2D data with added Gaussian noise, and 3D data with added Gaussian noise. The resulting maps are compared with the ground truth used for the simulation of the synthetic data.

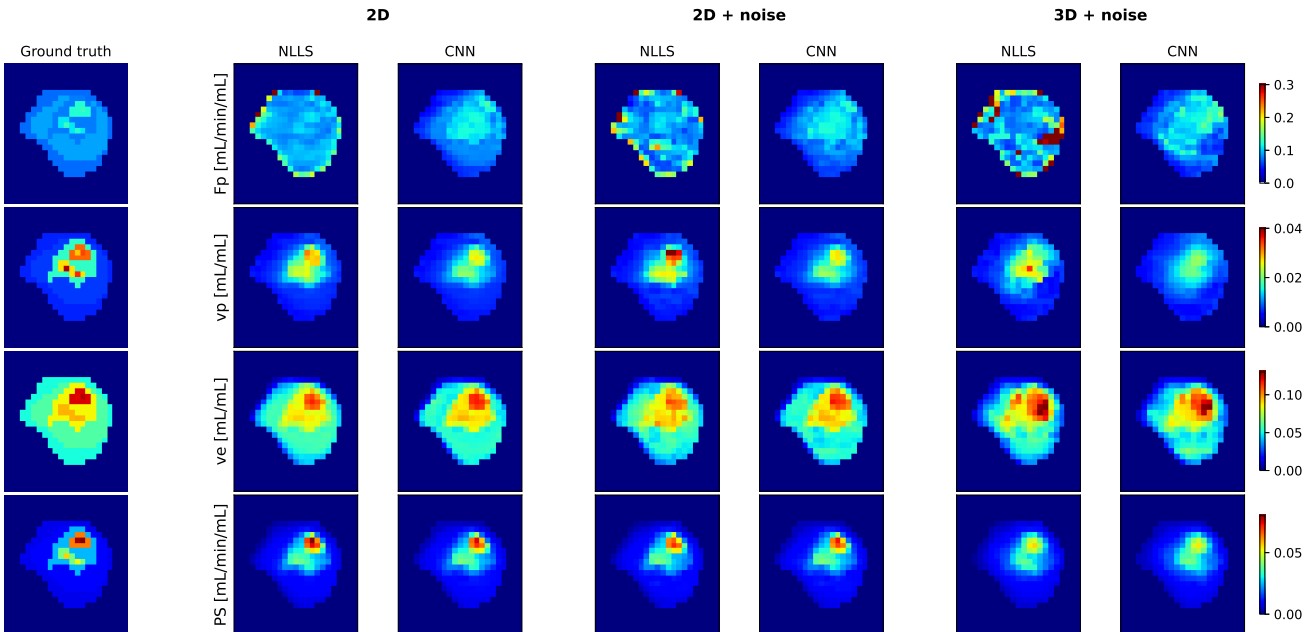

Fig. 4: Estimated parameters from the CNN and the NLLS method in the glioblastoma region for 2D data without noise, 2D data with added Gaussian noise, and 3D data with added Gaussian noise. The resulting maps are compared with the ground truth used for the simulation of the synthetic data.

The estimated perfusion maps from the datasets with different sampling periods are shown in Fig. 5. When the sampling period was prolonged to 2.4 s (4th and 5th column), the results were comparable with the estimates from the datasets with 1.2 s. For the longest sampling period, 4.8 s, (last two columns), the NLLS method with the 2CX model failed due to

the insufficient sampling necessary for the 2nd-generation PK models. In contrast, the CNN provided still correct parameter estimates corresponding to the ground truth, with a lower SD.

The parameters estimated in the glioblastoma are shown again in detail in Fig. 6. The results correspond with the previously observed phenomenons – the $F_p$ estimated by the

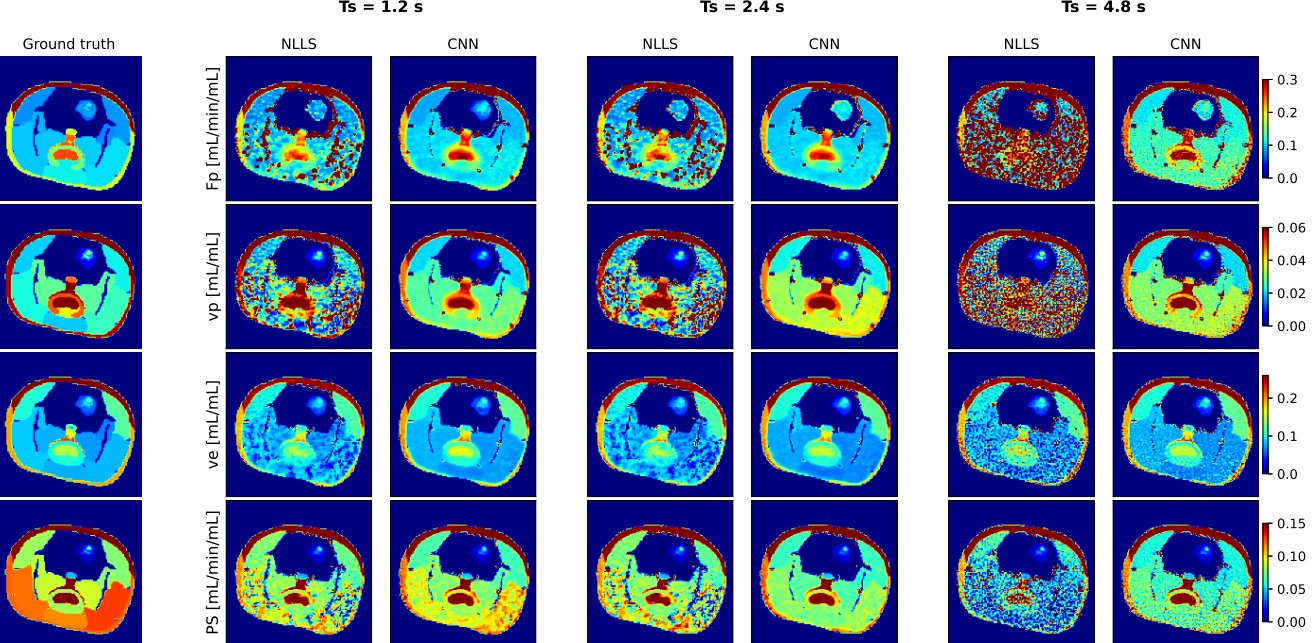

Fig. 5: Estimated parameters from the CNN and the NLLS method for 2D data with added Gaussian noise and three sampling periods, $T_s$ = 1.2 s, 2.4 s, and 4.8 s. The resulting maps are compared with the ground truth used for the simulation of the synthetic data.

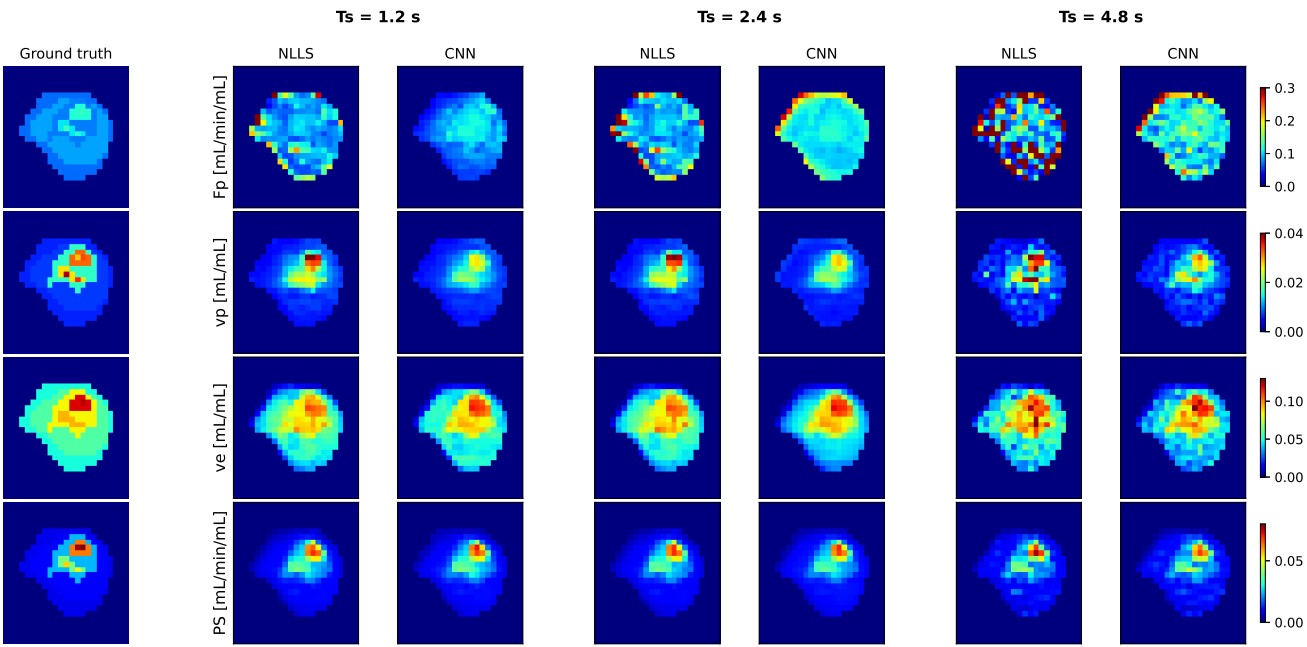

Fig. 6: Estimated parameters from the CNN and the NLLS method for 2D data in the glioblastoma region with added Gaussian noise and three sampling periods, $T_s$ = 1.2 s, 2.4 s, and 4.8 s. The resulting maps are compared with the ground truth used for the simulation of the synthetic data.

NLLS method was highly inaccurate for all the sampling periods; for the longest $T_s$, the results of the NLLS method were inaccurate due to the insufficient temporal resolution. For the remaining parameters and the $T_s$ = 1.2 s and 2.4 s, the NLLS method provided results with low deviation from the ground truth. The CNN provided quite stable results, also with low deviation from the ground truth; however, for the $F_p$, prolongation of the sampling period resulted in bias of the estimated parameter.

The error was again quantified with the MAPE and SD, and

TABLE I: MAPE and SD of the CNN-estimated parameters and NLLS-estimated parameters compared to the ground truth parameters for different acquisitions and noise levels.

Whole phantom

| MAPE [%] | 2D | | 2D with noise | | 3D with noise | |
|---|---|---|---|---|---|---|
| | NLLS | CNN | NLLS | CNN | NLLS | CNN |
| **Fp** | 383.6 ± 5529.6 | 23.0 ± 131.7 | 2002.9 ± 13104.5 | 25.1 ± 137.2 | 7055.5 ± 23265.9 | 36.4 ± 177.4 |
| **vp** | 17.5 ± 56.4 | 25.0 ± 45.3 | 55.4 ± 88.7 | 24.9 ± 44.4 | 90.1 ± 102.2 | 26.7 ± 36.8 |
| **ve** | 11.5 ± 24.2 | 11.6 ± 20.8 | 20.4 ± 32.8 | 12.1 ± 21.2 | 28.3 ± 43.6 | 14.9 ± 29.5 |
| **PS** | 12.0 ± 29.1 | 19.3 ± 20.2 | 28.3 ± 136.9 | 20.2 ± 22.4 | 44.2 ± 92.3 | 24.4 ± 24.0 |
| **BAT** | 4.1 ± 14.1 | 1.6 ± 2.9 | 4.7 ± 14.0 | 1.7 ± 2.9 | 6.4 ± 14.0 | 2.0 ± 3.0 |

Glioblastoma

| MAPE [%] | 2D | | 2D with noise | | 3D with noise | |
|---|---|---|---|---|---|---|
| | NLLS | CNN | NLLS | CNN | NLLS | CNN |
| **Fp** | 43.1 ± 138.9 | 20.7 ± 15.3 | 1524.8 ± 14518.5 | 21.1 ± 15.2 | 2066.4 ± 14376.2 | 27.0 ± 20.2 |
| **vp** | 19.8 ± 19.6 | 18.9 ± 19.6 | 21.4 ± 22.3 | 18.7 ± 16.6 | 26.1 ± 27.4 | 24.6 ± 20.0 |
| **ve** | 19.1 ± 21.7 | 20.0 ± 21.3 | 20.6 ± 21.6 | 21.2 ± 20.8 | 27.5 ± 21.6 | 28.0 ± 20.6 |
| **PS** | 24.6 ± 26.1 | 24.6 ± 24.4 | 24.7 ± 25.8 | 24.6 ± 24.4 | 28.0 ± 25.9 | 28.7 ± 26.2 |
| **BAT** | 2.3 ± 1.1 | 3.2 ± 2.1 | 2.7 ± 1.8 | 3.1 ± 2.1 | 3.0 ± 2.8 | 2.7 ± 1.9 |

TABLE II: MAPE and SD of the CNN-estimated parameters and NLLS-estimated parameters compared to the ground truth parameters for different sampling periods.

Whole phantom

| MAPE [%] | $T_s$ = 1.2 s | | $T_s$ = 2.4 s | | $T_s$ = 4.8 s | |
|---|---|---|---|---|---|---|
| | NLLS | CNN | NLLS | CNN | NLLS | CNN |
| **Fp** | 2002.9 ± 13104.5 | 25.1 ± 137.2 | 2641.9 ± 15408.9 | 24.3 ± 130.8 | 11915.6 ± 31847.5 | 49.8 ± 183.7 |
| **vp** | 55.4 ± 88.7 | 24.9 ± 44.4 | 57.6 ± 83.1 | 34.0 ± 49.6 | 110.4 ± 202.1 | 34.7 ± 108.3 |
| **ve** | 20.4 ± 32.8 | 12.1 ± 21.2 | 20.7 ± 27.6 | 13.4 ± 20.5 | 35.4 ± 74.9 | 17.3 ± 33.3 |
| **PS** | 28.3 ± 136.9 | 20.2 ± 22.4 | 27.7 ± 54.0 | 25.6 ± 20.2 | 67.4 ± 244.5 | 34.1 ± 57.4 |
| **BAT** | 4.7 ± 14.0 | 1.7 ± 2.9 | 5.9 ± 12.7 | 3.5 ± 2.4 | 10.2 ± 10.7 | 6.6 ±2.1 |

Glioblastoma

| MAPE [%] | $T_s$ = 1.2 s | | $T_s$ = 2.4 s | | $T_s$ = 4.8 s | |
|---|---|---|---|---|---|---|
| | NLLS | CNN | NLLS | CNN | NLLS | CNN |
| **Fp** | 1524.8 ± 14518.5 | 21.1 ± 15.2 | 813.7 ± 10292.8 | 57.1 ± 58.1 | 18414.4 ± 45365.9 | 68.4 ± 84.7 |
| **vp** | 21.4 ± 22.3 | 18.7 ± 16.6 | 21.5 ± 21.8 | 18.8 ± 16.4 | 30.7 ± 29.9 | 22.4 ± 19.0 |
| **ve** | 20.6 ± 21.6 | 21.2 ± 20.8 | 20.9 ± 21.6 | 28.4 ± 20.7 | 22.9 ± 21.9 | 27.1 ± 19.8 |
| **PS** | 24.7 ± 25.8 | 24.6 ± 24.4 | 24.7 ± 25.6 | 25.2 ± 23.7 | 30.4 ± 31.4 | 29.7 ± 25.7 |
| **BAT** | 2.7 ± 1.8 | 3.1 ± 2.1 | 4.7 ± 2.2 | 0.9 ± 0.7 | 10.7 ± 7.0 | 3.2 ± 1.4 |

is summarized in Table II. The values support all the above-mentioned observations.

### B. Real data

The parameter maps estimated by the NLLS method and the CNN in the mouse tumor are shown in Fig. 7. The tumor contained necrotic tissue (at the bottom in the middle), where the parameters could not be estimated reliably (because the PK model is not valid in tissue with no vasculature). In the remaining volume of the tumor, the CNN proved to be more robust to the noise than the NLLS as in the case of synthetic data, providing more visually consistent parameter maps.

### IV. DISCUSSION AND CONCLUSION

In contrast to the NLLS method, the CNN provided more robust results under all tested conditions. Thanks to the self-supervised training of the CNN, the model should be able to estimate the parameters similarly or even more reliably than the NLLS method from various types of datasets. An important factor is the computational time of the parameter estimation, which was reduced approximately 500× for our tested implementations of the CNN and the NLLS method (for the 2D dataset with noise, the NLLS fitting took 9 min 24 s, the CNN parameter prediction took 4.96 s; the NLLS fitting was parallelized with 64 workers). These results might be promising for the clinical application of quantitative DCE-MRI, introducing more reliable and faster solution with the DL approach.

However, we are aware of several factors that could influence the results of the CNN and that we aim to solve in our future work. First, the network was trained with a single AIF, which was also used for the NLLS data processing. In general, the AIF is examination-specific (although many implementations of DCE-MRI analysis assume a standard population-based AIF). The AIF selection has a significant influence on the parameter estimates resulting from the NLLS method. Hence the perfusion parameter estimates are expected to be inaccurate when the CNN has been trained with an AIF deviating from the true one. Therefore, this behaviour should be tested in the future, and possibly the network should be trained to accommodate for different AIFs. Alternatively, the AIF could be treated as an additional input to the NN. [6], [11], [14]

The network was successfully trained for different sampling periods, and it was able to estimate parameters accurately even

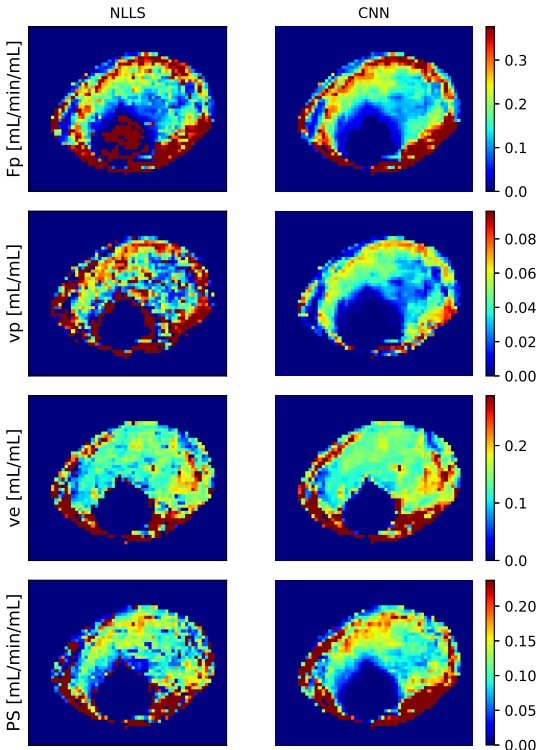

Fig. 7: Parameters maps estimated by NLLS and by CNN approach in the tumor area of a real dataset.

for the highest tested sampling period, $T_s$ = 4.8 s, when the NLLS method mostly failed. The processing of datasets with different sampling periods required retraining of the model. More general approach would be desirable, e.g. resampling of the signals before processing with the CNN. Furthermore, the acquisition time of the input sequence may vary. One of the possible solutions for that might be cropping of the signals to an estimated minimal signal length necessary for the CNN to predict the parameters accurately. Alternatively, another model, which is not dependent on signal length, might be implemented, e.g. one of the recurrent architectures such as GRU or LSTM. [5], [13]

In our work, we tested the model on simulated and preclinical data. Testing on clinical datasets is necessary for further development of this method. The AIFs differ significantly between small animals and humans, therefore, the incorporation of AIF information would be beneficial for the accuracy of the DL parameter estimates in clinical data. In addition, better results on various real datasets may be achieved by the incorporation of real data into the training dataset.

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
