# OpenReview forum: "Deep-Learning Estimation of Second-Generation Pharmacokinetic-Model Parameters in DCE-MRI"
_IEEE.org/EMBS/BHI/2024/Conference — IEEE BHI'24_

### Official Review · Reviewer_cqa7 · 2024-08-07
**Deep-Learning Estimation of Second-Generation Pharmacokinetic-Model Parameters in DCE-MRI**

**Overall Rating:** 7
**Confidence:** 3

**Other Quality Metrics:**

Clarity of writing: excellent
Clinical Significance: great
Methodological Novelty: good
Experiments and Results: great

**Questions For The Authors:**

If the global pathway has dilated convolutions, how are the authors ensuring that the output of the local and global paths is the same size for concatenation?

**Strengths:**

Description of the problem along with the technical gap are clear. It is clear what the authors are testing and how they are testing it.

Authors provided extensive details about their experimental setup. Experiments include simulated and real-world data.

Authors provided both quantitative and qualitative results of their experiments.

Authors provided a detailed description of the limitations of their proposed method.

**Summary Of The Paper:**

This study explored using a CNN to predict second-generation model parameters in dynamic contrast-enhanced MRI. The CNN was tested on simulated and real-world data with varying levels of noise and sample rates. Results showed that the CNN provided more stable and robust outcomes compared to traditional model fitting methods.

**Weaknesses:**

The authors state that their model is trained in an unsupervised fashion. However, it looks like the model is trained by regressing on the original concentration curve. It seems like the concentration curve works as a label in this situation.

Tables should include error bounds to show statistical significance.

---

### Official Review · Reviewer_Lfvg · 2024-08-10
**Deep-Learning Estimation of Second-Generation Pharmacokinetic-Model Parameters in DCE-MRI**

**Overall Rating:** 7
**Confidence:** 4

**Other Quality Metrics:**

a. Excellent

b. Good

c. Great

d. Excellent

**Questions For The Authors:**

1. Have you considered training the CNN with multiple AIFs to improve its robustness and applicability ?

2. Can you elaborate on potential challenges and necessary adjustments when applying the CNN model to human clinical datasets?

**Strengths:**

1. Their proposed approach addresses significant limitations of traditional pharmacokinetic (PK) models, particularly in terms of computational efficiency and noise sensitivity.

2. They perform extensive testing on simulated data with varying noise levels and real preclinical datasets, which enhances the credibility of their results.

**Summary Of The Paper:**

This paper explores the use of convolutional neural networks (CNNs) for estimating second-generation pharmacokinetic (PK) model parameters from Dynamic Contrast-Enhanced Magnetic Resonance Imaging (DCE-MRI) data. Traditional methods, primarily based on nonlinear least squares (NLLS) fitting, are computationally intensive and sensitive to noise, which limits their clinical applicability. By employing CNNs, the authors sought to provide faster and more robust parameter estimation, particularly under various noise conditions and sampling periods. The approach was validated using both simulated datasets and real preclinical data, and showed improved performance over conventional methods.

**Weaknesses:**

The network was trained using a single arterial input function (AIF), which means its applicability might be limited to cases where the AIF differs significantly.

---

### Decision · Program_Chairs · 2024-09-23

Accept